# Intramuscular Pulsed Radiofrequency Upregulates BNDF-TrKB Expression in the Spinal Cord in Rats as an Alternative Treatment for Complicated Pain

**DOI:** 10.3390/ijms25137199

**Published:** 2024-06-29

**Authors:** Cheng-Loong Liang, Cheng-Yo Yen, Hao-Kuang Wang, Yu-Duan Tsai, Cien-Leong Chye, Kuo-Wei Wang

**Affiliations:** 1Department of Neurosurgery, E-Da Hospital, I-Shou University, Kaohsiung City 82445, Taiwan; ed100183@edah.org.tw (C.-L.L.); ed101393@gmail.com (H.-K.W.); ed100184@edah.org.tw (Y.-D.T.); ed105482@edah.org.tw (C.-L.C.); 2Department of Orthopedic, E-Da Hospital, I-Shou University, Kaohsiung City 82445, Taiwan; ed100445@edah.org.tw; 3Department of Neurosurgery, E-Da Cancer Hospital, I-Shou University, Kaohsiung City 824005, Taiwan

**Keywords:** pulsed radiofrequency treatment, muscle, posterior spinal instrumentation, myofascial pain syndrome, brain-derived neurotrophic factor, vascular endothelial growth factor, intercellular adhesion molecule 1, tropomyosin receptor kinase B

## Abstract

Two cases of complicated pain exist: posterior screw fixation and myofascial pain. Intramuscular pulsed radiofrequency (PRF) may be an alternative treatment for such patients. This is a two-stage animal study. In the first stage, two muscle groups and two nerve groups were subdivided into a high-temperature group with PRF at 58 °C and a regular temperature with PRF at 42 °C in rats. In the second stage, two nerve injury groups were subdivided into nerve injury with PRF 42 °C on the sciatic nerve and muscle. Blood and spinal cord samples were collected. In the first stage, the immunohistochemical analysis showed that PRF upregulated brain-derived neurotrophic factor (BDNF) in the spinal cord in both groups of rats. In the second stage, the immunohistochemical analysis showed significant BDNF and tropomyosin receptor kinase B (TrkB) expression within the spinal cord after PRF in muscles and nerves after nerve injury. The blood biomarkers showed a significant increase in BDNF levels. PRF in the muscle in rats could upregulate BDNF-TrkB in the spinal cord, similar to PRF on the sciatica nerve for pain relief in rats. PRF could be considered clinically for patients with complicated pain and this study also demonstrated the role of BDNF in pain modulation. The optimal temperature for PRF was 42 °C.

## 1. Introduction

Pulsed radiofrequency (PRF) has been used for several years to treat various pain conditions: short bursts of radiofrequency delivered to the target nerve, producing an effect on signal transduction to reduce pain; this procedure does not produce a neural lesion, but neuromodulation [1]. It was originally designed to thermally coagulate nerve tissue, thereby blocking the nociceptive input [2]. We often encounter patients with a history of posterior instrumented spinal fusion and persistent pain near the surgical site. One potential cause of this pain is facet joint arthropathy at or adjacent to the level of fusion, and patients with posterior pedicle screws are associated with a risk of facet joint pain [3]. In patients without instrumentation, facet joint pain is often successfully treated with PRF in the medial branch nerve; however, patients with posterior spinal instrumented fusion and facet joint pain may be ineligible for PRF due to screws and postoperative adhesions [4]. In a recent study on chronic pain, some myofascial pain syndromes were classified as a mixed-type phenotype, such as nociceptive combined with nociplastic pain, which arises from altered nociception despite no clear evidence of actual or threatened tissue damage causing the activation of peripheral nociceptors or evidence of disease or lesion of the somatosensory system causing pain [5,6]. Brain-derived neurotrophic factor (BDNF) is a 12.4 kDa basic protein that was initially isolated from the pig brain and is widely expressed in the peripheral and central nervous systems. In the dorsal root ganglia (DRG), BDNF is expressed in small- and medium-sized neurons and is anterogradely transported to the central terminals of the spinal cord. Numerous studies have shown that BDNF modulates inflammatory and neuropathic pain. First, in formalin- and carrageenan-induced pain models, BDNF was upregulated in the DRG and spinal cord and sequestration of the upregulated BDNF reduced pain hypersensitivity [7,8,9]. Hence, in this study, owing to the absence of an adequate animal model for complicated pain, such as facet joint pain, after posterior spinal instrumentation and myofascial pain syndrome, we proposed a hypothesis of an alternative treatment for such complicated pain and directly applied PRF to the muscles of the lower limbs of rats to investigate the effect on muscles and the spinal cord and compared the effect with that of PRF on the sciatic nerve including the histopathological changes and biomarkers in the blood and spinal cord.

## 2. Results

### 2.1. The First Stage

In this stage, we evaluate the effects of PRF on the muscle and sciatic nerve at different temperatures including the BNDF expression in the spinal cord, biomarkers in the blood, and immunohistochemical staining in the muscle. Regarding the expression of BDNF in the spinal cord, the PRF groups demonstrated significant expression when compared with the sham group; however, no difference was observed between the 42 °C and 58 °C groups (Figure 1).

In the analysis of biomarkers in the blood, in the muscle groups, there was significant elevation in the 58 °C group (P trend < 0.001) but not in the 42 °C group (P trend = 0.313) across the time points in the analysis of BDNF levels. In addition, the elevation in the 58 °C groups was significantly greater than that in the 42 °C group (P for interaction = 0.035) (Figure 2A). In the SCN groups, there was significant elevation in the 58 °C (P trend = 0.004) and 42 °C groups (P trend = 0.022) across the time points. Noticeably, the elevation was significantly greater in the 58 °C group than in the 42 °C group (P for interaction = 0.03) (Figure 2B). No significant elevation in the ICAM-1 and VEGF levels in both the 42 °C and 58 °C groups across the time points was observed (Figure 2C, VEGF).

In the muscle nerve endings, the immunohistochemical staining (IHC) showed the expression of S100 protein and protein gene product 9.5 (PGP 9.5) (Figure 3).

In the IHC for the expression of BDNF and VEGF, both muscle groups showed BDNF and VEGF expression (Figure 3).

In the first stage, no neurological deficit was observed after these interventions.

### 2.2. The Second Stage

In this stage, we evaluated the effects of PRF on the muscle and nerve injury and we measured TrkB expression in the spinal cord. Regarding the expression of BDNF and TrkB in the spinal cord, the PRF groups demonstrated a significantly higher expression when compared with the control group; however, no difference was found between the nerve ligation and muscle groups (Figure 4).

Regarding the level of BDNF in the blood, the nerve injury group demonstrated significantly higher expression at all time points when compared to the control group (*p* < 0.05). The muscle group also exhibited higher expression at 10 days and 14 days when compared to the control group (*p* < 0.05). Noticeably, no significance difference between the nerve and muscle groups was observed at all time points (Figure 2D).

The von Frey test showed a significant pain reduction in these two groups when compared to the control group (Figure 5).

## 3. Discussion

Since its introduction by Sluijter in 1996, the use of PRF on the medial branches to treat facet joint pain has been practiced, and the results have been documented [10,11]. However, in some conditions, such as patients with metallic posterior spinal instrumentation and myofascial pain, the instrumentation causes adhesion and destruction of the facet joint, which is a difficult condition for PRF implementation. Myofascial pain is one of the most common causes of musculoskeletal pain in clinical practice and occurs in 30% of the patients in pain clinics [12]. Several studies have been conducted in this field using PRF; however, the clinical effects of this procedure, such as on the degree and duration of pain relief, and quality of life, have not been demonstrated. Most studies have demonstrated a pain reduction within weeks and months of PRF at trigger points [13,14]. Current animal models cannot provide the exact mechanism for such rapid pain relief because many mechanisms are involved, including increased or altered muscle demands; prolonged muscle contraction, such as postural inadequacies in the workplace, proximal nerve compression, and the resultant muscle spasm; and post-trauma conditions. [15]. In a recent study on chronic pain, myofascial pain syndrome was classified as mixed-type pain, including nociceptive and nociplastic pain, which may arise from altered nociception despite no clear evidence of actual or threatened tissue damage causing the activation of peripheral nociceptors, or evidence of disease or lesions of the somatosensory system causing pain [5,6]. Hence, we proposed our research on PRF in muscles, and investigated the effect of PRF on muscles and the possible mechanism for pain control. This study was divided into two stages to investigate the effect of PRF on muscles and to compare the effects of PRF on muscles and nerves. In the first stage, PRF was directly applied on the gastrocnemius muscles of rats and the effect of PRF within the muscles on the 7th day was investigated. We also applied PRF to the sciatic nerve and compared the effects of PRF on the muscle and sciatic nerve at the same stage. Among the biomarkers in the blood, a higher temperature of PRF caused higher BDNF levels, but with no elevation in ICAM-1 and VEGF levels, and no systemic inflammatory effects. Using IHC, we found that PRF increased the expression of S100, BDNF, ICAM-1, VEGF, and PGP9.5 in the muscle, especially in the high-temperature group. S100 protein has a damage-associated molecular pattern [16]. PGP9.5 immunostaining is the gold standard for diagnosing small-fiber neuropathy with hyperalgesia and allodynia [17]. ICAM-1 and VEGF are associated with inflammation [18]. From the biomarkers and IHC results, we found that PRF caused local inflammation in muscles and nerve reactions within the muscle, and higher temperatures caused more local inflammation in the muscle and increased BDNF levels in the blood. In the spinal cord, PRF caused the upregulation of BDNF at both temperatures, and no difference was found between the two different temperature groups. In the first stage, PRF on the muscle caused a significant elevation in BDNF levels in the blood, muscle, and spinal cord. BDNF is a small basic protein that is a member of the neurotrophin family of growth factors and is important for neural development, neurological function, and pain control [19]. The positive staining of PGP9.5 showed that PRF caused small-fiber neuropathy, and such small-diameter “pain” fiber activation was related to the elevation in spinal BDNF levels [20]. Elevated spinal levels of endogenous BDNF are related to pain behavior [21]. Following previous studies and our results, we demonstrated the role of BDNF in sensory plasticity, such as central sensitization, nociception, and pain, in the intact central nervous system [21], and that 42 °C PRF was enough to cause this effect.

In the second stage, we performed sciatic nerve ligation to mimic a nerve injury and applied 42 °C PRF in the nerve and muscle. The measurement of blood biomarkers showed a significant and progressive elevation in BDNF levels after PRF, especially in the nerve injury group, and PRF caused a progressive elevation in BDNF levels at days 10 and 14 in the muscle group. In the spinal cord, PRF upregulated BDNF expression in both groups. The neurological tests for pain also showed a significant effect in both groups. PRF on the muscle could provide a similar effect in pain reduction as PRF for nerve injuries and the associated factor was BNDF. In our previous study, we observed poor outcomes after PRF for facet joint pain in patients who underwent posterior metallic spinal instrumentation. In addition to the cause of lumbar spinal alignment, posterior screws and postoperative adhesions may also interfere with this outcome [22]. Intramuscular PRF may be an alternative treatment option. At this stage, we also checked the level of TrkB in the spinal cord and found that the elevation in the TrkB receptor was similar to that of BDNF. BDNF-TrkB signaling is associated with the modulation of pain perception, nociceptive plasticity, and spinal learning after cervical spinal cord injury [21,23,24]. A study of neuropathic pain has reported that the BDNF-TrkB pathway can significantly relieve neuropathic pain and is a potential target for the treatment of neuropathic pain [25]. After spinal cord injury, decreased BDNF and TrkB levels were observed in the acute stage, and elevated levels of BDNF and TrkB were observed between 3 and 6 weeks after the spinal cord injury. The BDNF-TrkB pathway is related to the recovery from a spinal cord injury [26]. In studies of myofascial pain, the treatment of myofascial trigger point pain (TrP) is reportedly used; however, the current literature on myofascial TrP has no consensus on its use as a diagnostic criterion, and no gold standard exists for its assessment [6]. Myofascial TrP is mainly considered as nociceptive pain [6]. It is important to note that the identification of the source of nociception does not exclude the possibility of concomitant nociplastic or neuropathic pain. Muscle TrP represents a peripheral source of nociception and may play an important role in the transition from localized pain to generalized pain conditions via the enhanced central sensitization and decreased descending inhibition [27]. It is possible that long-lasting and even short to moderate myofascial pain from nociceptive generators could facilitate central sensitization and, therefore, evolve into a nociplastic phenotype [28]. BDNF acts as a pain mediator (a factor that contributes to the initiation and development of pain) and modulator (a factor that regulates pain) and performs its biological functions through the TrkB receptor [28]. Our result demonstrated that intramuscular PRF could upregulate BDNF-TrkB in the spinal cord and cause a progressive elevation in BDNF levels in the blood, which was obvious at 14 days. From the perspective of psychiatric disease, there exists a strong correlation between myofascial pain and the conditions of depression, anxiety, and stress [28,29,30]. Increased BDNF levels could alleviate depression and anxiety [31]. In a recent study, it demonstrated musculoskeletal pain has a great impact on patients’ quality of life, since it might cause sleep interruptions, fatigue, depressed moods, activity limitations, and restrictions on activities, and it is even more disabling when the pain is related to sports or work. Although central nociceptive pathways contribute to degenerated pain, crosstalk between the immune system and nociceptive neurons is central to this pain [32]. From our results, BDNF was the associated biomarker and it was related to immune system. Intramuscular PRF should be considered as the treatment for myofascial pain (for the nociceptive, nociplastic pain) based on our results. No significant difference was observed in the expression of BDNF and TrkB within the spinal cord in these treatment groups and intramuscular PRF could provide the same effect. In our study, we checked many biomarkers in the blood and only BDNF was present in the blood and spinal cord. BDNF was considered as a clinical factor for pain. 

## 4. Materials and Methods

This was a two-stage animal study. All experiments were approved by the Institutional Animal Care (IACUC-105005) and Use Committee (IACUC) of E-DA Hospital and complied with the IACUC Guide for the Care and Use of Laboratory Animals. Adult male Sprague–Dawley rats (*n* = 30; weight: 250–300 g) were group-housed with a 12/12 h light–dark cycle and were provided a standard diet.

### 4.1. First Stage: PRF on the Muscle and Sciatic Nerve without Injury 

#### 4.1.1. PRF Treatment

Experimental protocol (first stage): In the first stage, the animals were divided into five groups: (1) sham group, skin incision only (*n* = 6); (2) muscle group with a high temperature of 58 °C, PRF at 58 °C (*n* = 6); (3) muscle group with 42 °C, PRF at 42 °C (*n* = 6); (4) sciatic nerve group with a high temperature of 58 °C, PRF at 58 °C (*n* = 6); and (5) sciatic nerve group with 42 °C, PRF at 42 °C (*n* = 6).

#### 4.1.2. Muscle Groups

Initially, rats were anesthetized with an intraperitoneal injection of pentobarbital sodium (50 mg/kg). After a toe-pinch test was performed to confirm that the rats were unconscious, a small skin incision was made above the gastrocnemius muscle, and the PRF electrode was inserted into the muscle. The groups were treated with a 2 Hz pulsed frequency current for 180 s at 42 °C and 58 °C, which was delivered through a radiofrequency generator (Neurotherm NT 2000IX, Abbott, Soma Tech Intl, Bloomfield, CT, USA).

#### 4.1.3. Sciatic Nerve Group 

After anesthesia, incisions were created above the biceps femoris, and the sciatic nerves (SNs) were exposed through blunt dissection [33]. The PRF electrode was then placed close to the SN. These groups were treated with a 2 Hz pulsed frequency current for 180 s at 42 °C and 58 °C, which was delivered through a radiofrequency generator (Neurotherm NT 2000IX Abbott, Soma Tech Intl, Bloomfield, CT, USA).

#### 4.1.4. Neurological Function

All animals performed well on the grid walk when tested within 24 h and 7 days postintervention, indicating that the intervention did not result in major sensory–motor impairment of the hind limbs.

#### 4.1.5. Determination of Intercellular Adhesion Molecule 1 (ICAM-1), Vascular Endothelial Growth Factor (VEGF), and BDNF Levels

Blood samples were collected in tubes containing potassium acetate before the injury and at selected time points after the injury (24 h, 48 h, 72 h, and 7 days). The samples were then centrifuged at 3000× *g* for 5 min, immediately frozen, and stored at −80 °C. ICAM-1, VEGF, and BDNF levels were measured using commercially available quantitative sandwich enzyme-linked immunosorbent assay (ELISA) kits (R&D Systems, Minneapolis, MN, USA).

#### 4.1.6. Histology and Immunohistochemistry

The experimental rats in each group were sacrificed 7 days after the PRF procedure. The rats were deeply anesthetized with isoflurane, and the heart was exposed. Each rat was perfused with 100 mL of saline, followed by phosphate buffer (0.1 mol/L, pH 7.4) with 4% paraformaldehyde (400 mL). The muscles and thoracic spinal cords (2 cm long; T8, T9, and T10) were removed, fixed in 10% formalin solution for 48 h, and embedded in paraffin wax. A series of transverse sections with a thickness of 5 μm were cut using an RM2000R microtome (Leica Microsystems, Wetzlar, Germany), positioned on slides, and placed in a drying oven overnight. The slides were then deparaffinized, rehydrated with decreasing ethanol passages, and stained either with hematoxylin and eosin or with an immunohistochemical marker, as described below. 

For hematoxylin and eosin staining, the slides were immersed in 0.1% hematoxylin (Ciba, Basle, Switzerland) for 10 min, washed in tap water for 15 min, immersed in 0.1% eosin (Ciba) for 5 min, and washed in distilled water. The sections were then dehydrated using ascending ethanol passages and mounted with DPX (Fluka, Honeywell, New Taipei City, Taiwan).

For immunohistochemical staining, the sections were incubated for 10 min in 3% hydrogen peroxide to block endogenous peroxidase activity. High-temperature antigen retrieval was performed in 10 mmol/L boiled citrate buffer (pH 6.0) for 20 min. Tissue sections were incubated overnight at 4 °C with mouse anti-BDNF immunoglobulin (Ig) G (diluted 1:200; Abcam, Cambridge, UK), rabbit anti-S100 beta IgG (diluted 1:300; Abcam, UK), mouse anti-PGP9.5 IgG (diluted 1:200; Abcam, UK), or mouse anti-VEGF IgG (diluted 1:100; Santa Cruz, CA, USA) in phosphate-buffered saline (PBS) containing 0.3% (*v/v*) Triton X-100, followed by incubation with EnVision (Zymed, CA, USA) solution at 37 °C for 30 min. Finally, the sections were incubated with the peroxidase substrate diaminobenzidine until the desired staining intensity was obtained, followed by slight counterstaining with hematoxylin, dehydration, and cover-slipping with Permount. Between incubations, the tissues were washed three times with PBS for 10 min each. Images were obtained using a microscope (Olympus BX43F, Tokyo, Japan) and the ProgRes CapturePro 2.8 Software (JENOPTIK, Augsburg, Germany). We analyzed the stained area in five random sections from each sample to obtain the mean ± standard deviation to verify significant differences between the groups.

### 4.2. Second Stage: PRF on Muscle and Sciatic Nerve after Nerve Injury

#### 4.2.1. Experimental Protocol (The Second Stage)

In the second stage, a nerve injury was inflicted as a sciatic nerve ligation [33]. The animals were divided into four groups: (1) sham group, skin incision-only (*n* = 6); (2) sciatic nerve ligation group (control group) (*n* = 6); (3) sciatic nerve ligation group at a temperature of 42 °C, PRF at 42 °C within the muscle (*n* = 6); and (4) sciatic nerve ligation group at a temperature of 42 °C, PRF at 42 °C on the proximal sciatic nerve (*n* = 6) [33].

#### 4.2.2. Neurological Function for Pain 

Pain cannot be directly measured in rats, but many methods that quantify “pain-like” behaviors or nociception have been developed. We used the manual von Frey test to assess pain behavior. The manual Von Frey test, developed by the physiologist Maximilian von Frey, is a method of evaluating mechanical allodynia in mice and rats. In this test, animals are placed individually in a small cage with a mesh or penetrable bottom. A monofilament is applied perpendicularly to the plantar surface of the hind paw until it buckles, delivering a constant pre-determined force (5.9–98 mN for rats) for 2–5 s. A response is considered positive if the animal exhibits any nocifensive behaviors, including brisk paw withdrawal, licking, or shaking of the paw, either during the application of the stimulus or immediately after the filament is removed. The plantar surface of the hind paw is the most commonly used area for testing. All animals were tested before and after the intervention and the test area was the plantar surface of the hind paw in a small cage with a mesh [34].

#### 4.2.3. BDNF Determination 

Blood samples were collected in tubes containing potassium acetate before the injury and at selected time points after the injury (24 h, 48 h, 72 h, and 14 days). The samples were then centrifuged at 3000× *g* for 5 min, immediately frozen, and stored at −80 °C. The BDNF levels were measured using commercially available quantitative sandwich ELISA kits (R&D Systems, Minneapolis, MN, USA).

#### 4.2.4. Histology and Immunohistochemistry

The experimental rats in each group were sacrificed 14 days after the PRF procedure, and the procedures performed were the same as those described in Section 4.1.6. We performed immunohistochemistry for BDNF and tropomyosin receptor kinase B (TrkB).

### 4.3. Statistical Analysis

#### 4.3.1. The First Stage

To determine the statistical significance of the BDNF expression in the fractions of the spinal cord, one-way analysis of variance (ANOVA) followed by the post Tukey test was used with the Bonferroni correction. Differences were considered statistically significant at *p* < 0.05.

Furthermore, the elevation in the values of the outcomes of interest (i.e., BDNF expression and ICAM-1 levels in muscle group or sciatic nerve group) across the time points was tested using a generalized estimating equation in which time was considered a continuous variable. The generalized estimating equation was calculated separately for the 42 °C and 58 °C groups. The extent of elevation between the 42 °C and 58 °C groups was compared using a generalized estimating equation (GEE), which included a two-way interaction effect of time by group (42 °C vs. 58 °C).

#### 4.3.2. The Second Stage

To determine the statistical significance of the BDNF and tropomyosin receptor kinase B (TrkB) expression in the fractions of the spinal cord, one-way ANOVA followed by the post Tukey test was used with the Bonferroni correction. Differences were considered statistically significant at *p* < 0.05. We further were interested in comparing the BDNF expression in the blood among the groups (nerve, muscle, and no treatment), including the comparison between any two groups at a specific time point and the difference between baseline and a later time point within one group (e.g., nerve). The results were obtained by using the GEE model by including the interaction effect between groups and time points in which time was considered a categorical variable. The above-mentioned comparisons were acquired from the simple main effect of the interaction. Data analyses were conducted using SPSS 25 (IBM SPSS Inc., Chicago, IL, USA).

## 5. Conclusions

The 42 °C PRF procedure in the muscle caused elevated expression of BDNF-TrkB in the blood and spinal cord, which could provide an effect similar to that of regular PRF in treating nerve injuries in rat. It may be considered clinically for patients undergoing posterior spinal instrumentation for facet joint pain and myofascial pain. In our study, BDNF was considered a clinical factor for pain.

## Figures and Tables

**Figure 1 ijms-25-07199-f001:**
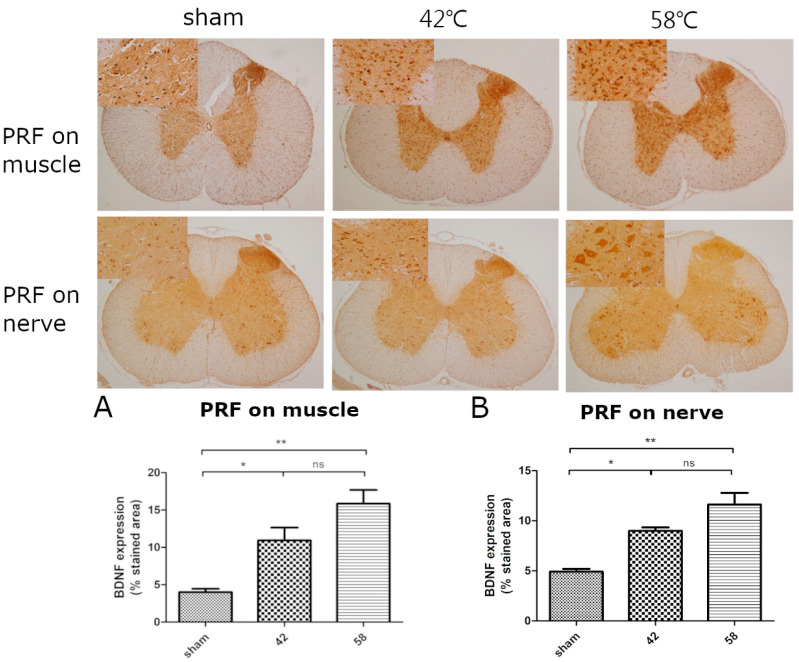
BDNF expression in spinal cord after pulsed radiofrequency on muscle and sciatic nerve. (**A**) Increased BDNF expression in the gray matter of spinal cord after PRF on muscle. The rats underwent a PRF procedure on the muscles of lower limbs at 42 °C or 58 °C or a sham operation; 40× and 400× magnification. Quantification of BDNF-positive stained area in each group. One-way ANOVA followed by post Tukey test was used for statistical analysis to compare control and treated groups. * *p* < 0.05 for 42 °C RF result that was significantly different from control and ** *p* < 0.01 for 58 °C RF result that was significantly different from control treatments. (**B**) BDNF expression in spinal cord after pulsed frequency on sciatica nerve. The rats underwent a PRF procedure on the sciatic nerve at 42 °C or 58 °C or a sham operation; 40× and 400× magnification. Comparison of the percentage of BDNF-positive stained area between groups. One-way ANOVA followed by post Tukey test was used for statistical analysis to compare control and treated groups. * *p* < 0.05 for 42 °C RF result that was significantly different from control and ** *p* < 0.01 for 58 °C RF result that was significantly different control treatments.

**Figure 2 ijms-25-07199-f002:**
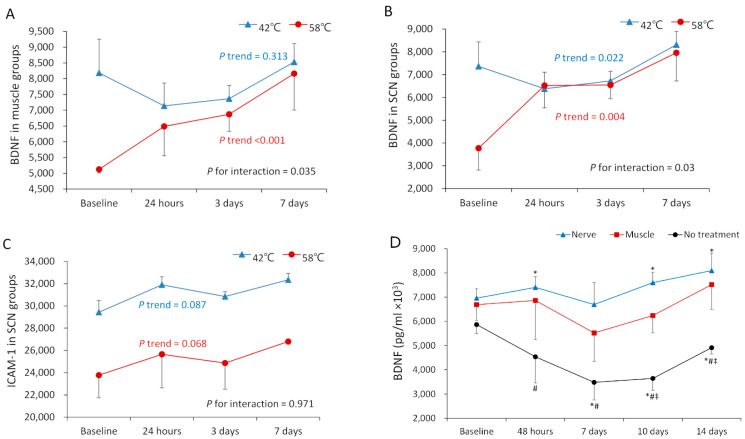
Analysis of BDNF and ICAM-1 levels in blood (in the first and second stages). (**A**) There was a significant elevation in BDNF levels in both groups in the muscle groups. (**B**) There was a significant elevation in BDNF levels in both groups in the SCN groups. (**C**) There was no significant elevation in ICAM-1 levels in both groups. (**D**) The nerve injury group demonstrated significantly different expression levels at all time points when compared to the control group (*p* < 0.05, asterisk “*”). The muscle group also exhibited higher expression at 10 days and 14 days when compared to the control group (*p* < 0.05, symbel “#≠”). Noticeably, no significance difference between the nerve and muscle groups was observed at all time points. * Indicates *p* < 0.05 verus basline within one group. # Indicates *p* < 0.05 between nerve and no treatment groups at the time point. ≠ Indicates *p* < 0.05 between muscle and no treatment groups at the time point.

**Figure 3 ijms-25-07199-f003:**
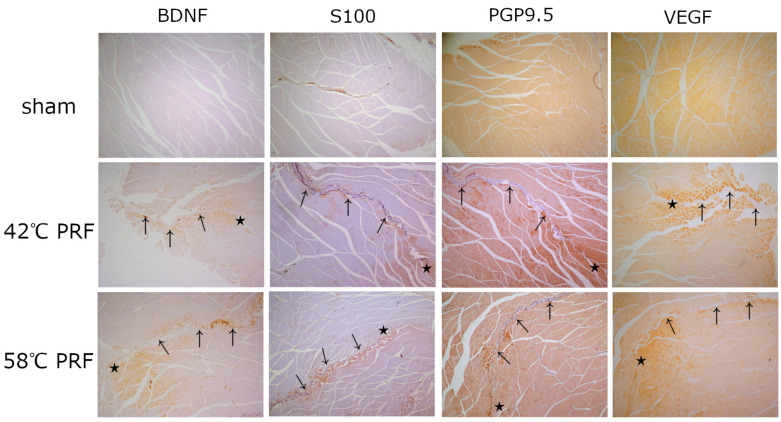
Immunohistochemistry of BDNF, S100, PGP9.5, and VEGF expression in muscle after pulsed radiofrequency at different temperatures. Seven days after PRF, the ablated muscles were harvested and processed for immunohistochemistry. Microscopic images show the elevation in BDNF, S100, PGP9.5, and VEGF expression around the probe (40× magnification). The arrows (↑) show the path of the PRF probe and the star (★) shows the tip of the probe.

**Figure 4 ijms-25-07199-f004:**
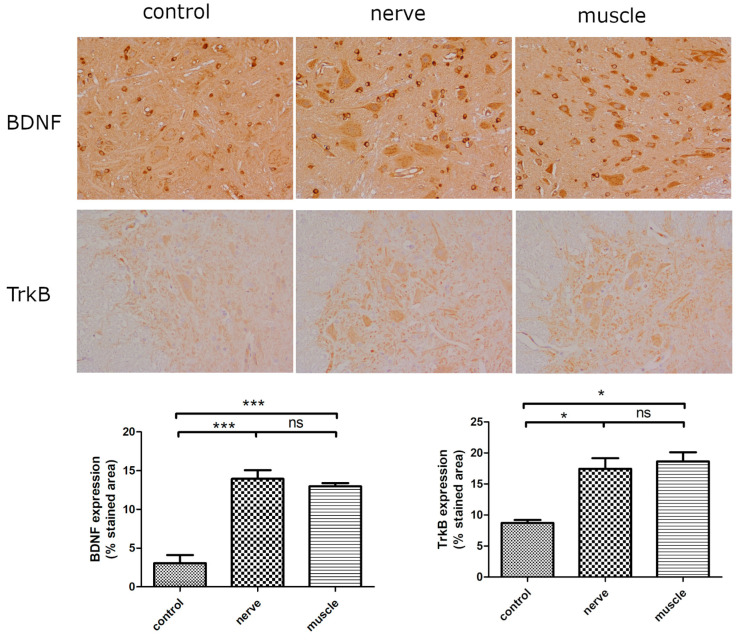
BDNF and TrkB expression in spinal cord after pulsed radiofrequency in muscle in the second stage. Increased BDNF expression in the gray matter of spinal cord after PRF. The rats underwent PRF procedure at 42 °C on the muscles of lower limbs or on nerve , 40× and 400× magnification. Quantification of BDNF stained area (400×) are expressed as mean ± SEM. *** *p* < 0.001 compared with control group (one-way ANOVA followed by Tukey-Kramer post hoc test). Increased TrkB expression in the gray matter of spinal cord after PRF. The rats underwent PRF procedure at 42 °C on the muscles of lower limbs or on nerve. Quantification of TrkB stained area (400×) are expressed as mean ± SEM. * *p* < 0.05 compared with control group (one-way ANOVA followed by Tukey-Kramer post hoc test).

**Figure 5 ijms-25-07199-f005:**
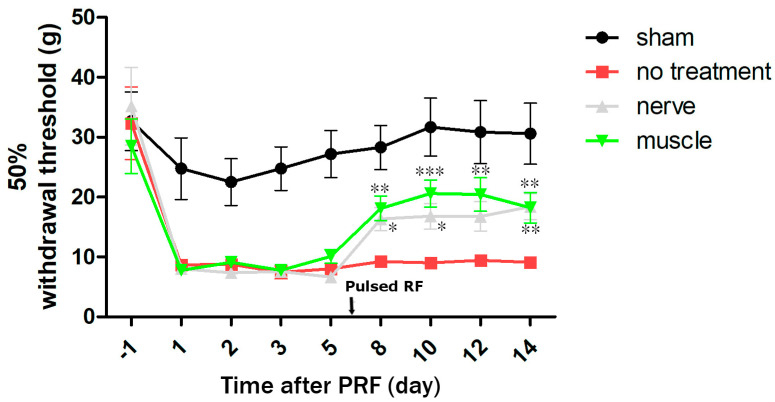
Von Frey test for pain behavior. The von Frey test showed a significant pain reduction in the PRF groups when comparing to the control group ( * *p* < 0.05, ** *p* < 0.01, *** *p* < 0.001).

## Data Availability

We ensured that the datasets were either deposited in publicly available repositories or presented in the main manuscript.

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
