# Peer review of "Intramuscular Pulsed Radiofrequency Upregulates BNDF-TrKB Expression in the Spinal Cord in Rats as an Alternative Treatment for Complicated Pain"

_ijms, 2024, doi:10.3390/ijms25137199_

Round 1

Reviewer 1 Report

Comments and Suggestions for Authors

I apologize for making a comment on the introduction in a language other than English. The commentary in the correct language is listed below.

The information contained in the introduction is insufficient. If the work is to be original it should include an introduction, the purpose of the work, material and methods, results, and conclusions.

It is not clear what objects are subject to observation, humans or animals.

What is a muscle or nerve group? The temperature of 58 degrees denatures the protein, it cannot be effective. It is not necessary to do research for this. The use of deep heat therapy for metal anastomoses is contraindicated.

Where did the PRF source come from and what specific parameters does it have; length, frequency, waveform, power?

To whom were the spinal cord samples taken?

The comments above apply to the summary.

The purpose of the work was not stated.

The summary and introduction do not indicate that the therapy is about a technique used in heat stimulation - 42 degrees and above 58 degrees, i.e. thermolesion.

The two methods of PRF application are incomparable, as they serve different purposes and have different tasks in medical practice. By definition, their effects must differ, so I consider the study pointless. Especially since the treatments mentioned are usually applied either to the articular surfaces of the interarticular joints or to the nerves. They certainly do not serve to treat myofascial pain.

The methodology of the paper is described chaotically, and the conclusions drawn do not serve the practical application of PRF. Transferring the results obtained under laboratory conditions on animals to human treatment is too far ahead, especially since the relevant methodology of the procedure on humans is not described.

Due to the above comments, the article is not eligible for publication without extensive revision.

Reviewer 2 Report

Comments and Suggestions for Authors

This paper is interesting since it deals with a potential new application for radiofrequency in the treatment for complicated pain.

Nevertheless, many concerns have to be addressed.

The title is long and not concise and incisive. You should also clearly express in the title the study model and the population on which it was carried out.

Keywords: I suggest to use MeSH terms.

Abstract: also in this case, the study model is not clearly presented, rats are not mentioned and the methods are confusing. Please correct it. 

Introduction: you should enlarge this section both for explaining why you chose the procedures you use and, overall, clearly stating the end points: these latter are not clear and do not coincide with those expressed in the materials section.

Results: they are well described.

Discussion: this section should be largely enriched. You should first of all explain which are the traditional and the new applications of radiofrequency in the treatment of musculoskeletal pain. To do that, I suggest the following reference:

-Farì, G., de Sire, A., Fallea, C., Albano, M., Grossi, G., Bettoni, E., Di Paolo, S., Agostini, F., Bernetti, A., Puntillo, F., & Mariconda, C. (2022). Efficacy of Radiofrequency as Therapy and Diagnostic Support in the Management of Musculoskeletal Pain: A Systematic Review and Meta-Analysis. Diagnostics (Basel, Switzerland)12(3), 600. https://doi.org/10.3390/diagnostics12030600

Materials: this section should be drastically improved.

It is necessary to describe the study model (is it a cross-sectional one? is it a randomized trial?) and the study populations (is it just rats?). Then, a sample size calculation is needed.

Authors contribution: the precise suddivision of the role and contribution of each authos should be stated.

Best regards and good luck 

Comments on the Quality of English Language

a Minor global revision is needed.

Round 2

Reviewer 1 Report

Comments and Suggestions for Authors

The authors followed the reviewers' comments. The article in its current form is eligible for publication.

Reviewer 2 Report

Comments and Suggestions for Authors

Thank you for the efforts to improve Your paper following my comments. It is now well structured, so no further corrections are needed in my opinion